# Non-Pathogenic Mopeia Virus Induces More Robust Activation of Plasmacytoid Dendritic Cells than Lassa Virus

**DOI:** 10.3390/v11030287

**Published:** 2019-03-21

**Authors:** Justine Schaeffer, Stéphanie Reynard, Xavier Carnec, Natalia Pietrosemoli, Marie-Agnès Dillies, Sylvain Baize

**Affiliations:** 1Unité de Biologie des Infections Virales Emergentes, Institut Pasteur, 69007 Lyon, France; schaeffer.justine.91@gmail.com (J.S.); stephanie.reynard@pasteur.fr (S.R.); xavier.carnec@pasteur.fr (X.C.); 2Centre International de Recherche en Infectiologie (INSERM, CNRS, ENS Lyon, Université Lyon I), 69007 Lyon, France; 3Hub de Bioinformatique et Biostatistique–C3BI, Institut Pasteur, USR 3756 CNRS, 75015 Paris, France; natalia.pietrosemoli@pasteur.fr (N.P.); marie-agnes.dillies@pasteur.fr (M.-A.D.)

**Keywords:** Lassa virus, Mopeia virus, viral haemorrhagic fever, plasmacytoid dendritic cells, type I interferon

## Abstract

Lassa virus (LASV) causes a viral haemorrhagic fever in humans and is a major public health concern in West Africa. An efficient immune response to LASV appears to rely on type I interferon (IFN-I) production and T-cell activation. We evaluated the response of plasmacytoid dendritic cells (pDC) to LASV, as they are an important and early source of IFN-I. We compared the response of primary human pDCs to LASV and Mopeia virus (MOPV), which is very closely related to LASV, but non-pathogenic. We showed that pDCs are not productively infected by either MOPV or LASV, but produce IFN-I. However, the activation of pDCs was more robust in response to MOPV than LASV. In vivo, pDC activation may support the control of viral replication through IFN-I production, but also improve the induction of a global immune response. Therefore, pDC activation could play a role in the control of LASV infection.

## 1. Introduction

Lassa virus (LASV) has been listed by the World Health Organisation as one of the emerging pathogens likely to cause severe outbreaks in the near future and for which few or no medical countermeasures exist [1]. In humans, it causes a viral haemorrhagic fever (VHF), Lassa fever (LF). LF is endemic in West Africa and causes tens of thousands of cases and several thousand deaths every year [2]. There is currently no approved vaccine against LASV and only one partially efficient treatment, ribavirin [3].

LASV is an old world arenavirus. This phylogenetic group includes highly pathogenic viruses, such as LASV or Lujo virus, and non-pathogenic viruses, such as Mopeia virus (MOPV) [4]. MOPV is phylogenetically very close to LASV and has been isolated from *Mastomys natalensis*, which is a LASV reservoir [5]. However, MOPV is non-pathogenic in non-human primates (NHPs), and no human case of MOPV infection has ever been reported [4]. Comparing MOPV and LASV, two very similar viruses with different pathogenic potential, may be a fruitful approach to identify immune and viral features involved in LASV pathogenesis. Fatal LASV infection in human is associated with immunosuppression, leading to uncontrolled viral replication and severe symptoms. In vivo studies have also shown that the survival of LASV-infected NHPs correlates with an early type I interferon (IFN-I) response and robust T-cell responses [6].

Plasmacytoid dendritic cells (pDC) are highly potent IFN-I producers [7]. They can detect viral infection through Toll-like receptors (TLR) 7 and 9, as well as RIG-I-like receptors (RLR). The constitutive expression of IRF7, which is usually an IFN-stimulated gene (ISG), partially explains the efficiency of IFN-I production by pDCs [7]. During lymphocytic choriomeningitis virus (LCMV) infection, pDCs are the major source of IFN-I [8]. In mice, pDCs were shown to be permissive to LCMV infection, and both infected and non-infected pDCs produced IFN-I [9]. Considering these results and the major role of IFN-I in LF, we decided to study the response of pDCs to LASV. We compared LASV and MOPV to evaluate the importance of the pDC response in the high pathogenicity of LASV.

The results presented here were obtained using primary human pDCs purified from the blood of healthy donors. pDCs were not productively infected by either MOPV or LASV. On the contrary, coculture between pDCs and MOPV- or LASV-infected cells led to the detection of viral proteins in pDCs. We then analysed the IFN-I response of pDCs to MOPV and LASV. Both MOPV- and LASV-infected pDCs produced IFN-I. However, the response of pDCs to LASV was rapidly shut down. A larger scale approach, using transcriptomic and multiplex protein detection, showed stronger activation of MOPV-infected than LASV-infected pDCs. Overall, these results show that MOPV is a better stimulus for pDC activation than LASV. As pDCs are involved in the very early IFN-I response in vivo, their ability to detect LASV and initiate a response could be critical during LF.

## 2. Materials and Methods

### 2.1. Virus and Cells

VeroE6 cells were grown in DMEM with 0.5% penicillin-streptomycin (PS) and 5% foetal bovine serum (FBS, all from Invitrogen, Cergy-Pontoise, France). Mopeia (AN21366 strain [4]) and Lassa (AV strain [10]) viruses were grown in VeroE6 cells at 37 °C, with 5% CO_2_. LASV and MOPV titres were determined by plaque immunoassays, as previously described [11]. MOPV and LASV with a FLAG-tagged Z protein (MOPV-Zflag and LASV-Zflag) were obtained by reverse genetics, as previously described [12]. Experiments with LASV were carried out in biosafety level 4 facilities (Laboratoire P4 Jean Merieux-Inserm, Lyon, France).

### 2.2. Cell Purification

Human peripheral blood was obtained from healthy donors with informed consent and was provided by the Etablissement Français du Sang (Lyon, France, agreement PLER/1-1820-05/05/14). Written informed consent was provided by all study participants. PBMCs were isolated by Ficoll (GE Healthcare, Velizy, France) centrifugation. pDCs were isolated using the Diamond Plasmacytoid Dendritic Cell Isolation kit II (Miltenyi Biotech, Paris, France). pDCs were cultured in RPMI 1640 Glutamax I, 0.5% PS, 10 mM HEPES, 1% nonessential amino acids, and 10% FBS (all from Invitrogen). R848 (Invitrogen) treatment at 1 µg/mL was used as a positive control of activation.

### 2.3. RT-qPCR

Cellular RNA was purified using the RNeasy kit (Qiagen, Courtaboeuf, France), followed by DNAse I (Qiagen) and Ambion DNAse (Thermo Fisher Scientific, Waltham, MA, USA) digestion. For viral RNA quantification, cells were treated prior to extraction with 0.05% trypsin (Invitrogen) to remove adsorbed virions. Viral RNA from the cell medium was purified using the QIAmp Viral RNA Mini Kit (Qiagen). IFN-I mRNA and viral nucleoprotein (NP) RNA was quantified by RT-qPCR, as previously described [11]. Relative IFN-I mRNA levels were calculated as 2^−ΔCt^, with Ct the cycle threshold and ΔCt = [gene Ct] − [GAPDH Ct]. NP RNA was quantified (by copy number) by comparing our samples with sequential dilutions of RNA standards. All runs were performed in duplicate using a LightCycler480 (Roche Diagnostics, Meylan, France).

### 2.4. Flow Cytometry

For the “vero” condition, VeroE6 cells were infected with MOPV-Zflag or LASV-Zflag (MOI = 0.3) and analysed 48 hours post infection (hpi) For the “pDC” condition, pDC were infected with MOPV-Zflag or LASV-Zflag (MOI = 0.1) and analysed 24 hpi. For the “coC” condition, VeroE6 cells were infected with MOPV-Zflag or LASV-Zflag (MOI = 0.3), pDCs were added 24 hpi and cells were analysed 48 hpi. For analysis, cells were stained with Lin1-FITC (BD Biosciences, Le-Pont-de-Claix, France) and CD303 (AC144)-PE-Vio770 (Miltenyi Biotech). Viral Z proteins were stained using the FoxP3 Staining Buffer Set, FcR Blocking Reagent, and human and anti-DYKDDDDK-APC (Miltenyi Biotech). Fluorescence was measured using a Gallios flow cytometer (Beckman Coulter, Brea, CA, USA) and analysed using Kaluza software version 1.2 (Beckman Coulter). pDCs were gated as Lin1-/CD303+ cells (Appendix A).

### 2.5. Transcriptomic Analysis

pDCs were infected for 12 h at a MOI = 1 with LASV, MOPV, or remained uninfected. Cellular RNA was purified using the RNeasy kit (Qiagen), followed by DNAse I (Qiagen) and DNAse (Ambion) digestion. Sequencing were performed by ViroScan3D (Lyon, France). RNA quality was checked using QuantiFluor RNA System (Promega, Charbonnières-les-Bains, France) and RNA 6000 Pico Kit (Agilent, Santa Clara, CA, USA). cDNA were synthesized using random priming of poly-A RNAs (NEXTFLEX Rapid Directional RNA-Seq Library Prep Kit, PerkinElmer, Boston, MA, USA). Single-end, 75-bp read-length NextSeq 500 High throughput sequencing of the cDNA library was performed. After demultiplexing and trimming of the adaptors (with Bcl2fastq), 30 million reads per sample were obtained. Sequencing quality was assessed for each sample (before and after mapping) using FastQC. Reads were aligned on the human genome (Human GRCh38.p7, from ENSEMBL) using STAR and a maximum mismatch rate of 5%. Reads aligned on each gene were counted using the module feature count. Statistical analysis of the read counts (quality checks, normalisation using scaling factors, fold change and *p*-values calculations) were performed using the R package SARTools (DESeq2) [13]. Genes were differentially expressed for *p* < 0.05. Heatmaps were generated with R (package heatmap2), using genes with differential expression for at least one pairwise comparison.

### 2.6. Luminex

pDCs were harvested 16 hpi and the culture medium collected. Fifty cytokines were quantified using the Milliplex map kit Human Cytokine/Chemokine Magnetic Bead Panel (PX38) and Human Cytokine/Chemokine Magnetic Bead Panel IV (Merck Millipore, Guyancourt, France). Runs were performed with a Magpix luminex (Merck Millipore).

### 2.7. Statistical Analysis

The mean and standard error of the mean (SEM) for each set of data were calculated using R. Graphs were generated and statistical analysis performed using SigmaPlot (SyStat Software Inc, San Jose, CA, USA). Differences were considered statistically significant at *p* < 0.05. Heatmaps were generated using R.

## 3. Results

### 3.1. pDCs Were Not Productively Infected by MOPV or LASV

We infected primary human pDCs with MOPV or LASV and quantified infectious particles or viral RNA in the culture supernatant (Figure 1a,b). The viral titres of both MOPV and LASV decreased over time, showing no release of infectious particles. At the same time, viral RNA levels in the supernatant did not increase. These results show that pDC infection by MOPV or LASV is not productive. To assess pDC infection earlier in the viral cycle, we quantified viral RNA from MOPV- and LASV-infected pDCs (Figure 1c). Adsorbed viral particles were removed by incubation with trypsin. There were no significant differences in the levels of MOPV and LASV RNA between 1 hpi, 1 day post-infection (dpi), and 2 dpi. Thus, there was no detectable replication and transcription of MOPV and LASV in pDCs.

In vivo, pDCs are not only in contact with viral particles, but also with infected cells. Infected cells could affect the properties of pDCs, including their permissivity. We tested whether LASV and MOPV behaved similarly by infecting VeroE6 cells with MOPV or LASV, culturing pDCs with viral particles or infected cells, and staining the pDCs for the viral Z protein (Appendix A). No Z-positive pDCs were found following incubation with viral particles (Figure 1d–f). However, following coculture with infected VeroE6, 10% (for LASV) to 30% (for MOPV) of pDCs were Z-positive. Z proteins detected in pDCs could either come from the neosynthesis or internalization of existing viral components. We also observed a decrease in infected VeroE6 in the pDC/infected VeroE6 cocultures (Figure 1e–g), suggesting that coculture with pDCs partially controlled VeroE6 infection by MOPV and LASV. VeroE6 cells are unable to produce IFN-I, but are susceptible to exogenous IFN-I, while pDCs are specialized in the production of IFN-I in response to pathogens. Thus, the control of VeroE6 infection observed in VeroE6-pDC cocultures could result from the production of IFN-I by pDCs.

### 3.2. The IFN-I Response of MOPV-Infected pDCs Lasts Longer Than That Induced by LASV

pDCs are highly potent IFN-I producers and are involved in the host response to arenaviruses. Accordingly, we quantified IFN-I production by MOPV- and LASV-infected pDCs. We first evaluated the time course of the IFN-I response to MOPV (Figure 2a). The IFNα1, IFNα2, and IFNβ genes were up-regulated in MOPV-infected pDCs as early as 6 hpi. The IFN-I response peaked at 12 hpi and then slowly decreased. We then chose an early timepoint (7 hpi) and a later timepoint (16 hpi) to compare the pDC response to MOPV and LASV. The IFNα1, IFNα2 and IFNβ mRNAs were up-regulated in both MOPV- and LASV-infected pDCs relative to uninfected pDCs at 7 hpi (Figure 2b–d). However, only MOPV-infected pDCs showed an over-expression of IFN-I mRNA at 16 hpi. pDCs were thus able to produce an IFN-I in response to both MOPV and LASV, but their response to MOPV was long-lasting.

### 3.3. The Global Activation State of pDCs Is Lower during LASV Than MOPV Infection

We further evaluated the pDC response to MOPV and LASV by quantifying the cytokines and chemokines in the culture supernatant of pDCs, 16 hpi (Figure 3a, Appendix A). IFNα2 was produced by MOPV- and LASV-infected pDCs, but in slightly lower amounts by cells infected with LASV. In contrast, IFNβ protein was not detected, neither for MOPV or LASV. This discrepancy with the results obtained by RT-qPCR (Figure 2) may originate from the post-transcriptional regulation of IFNβ mRNAs or the uptake and recycling of IFNβ. We also observed IP-10 production in both MOPV and LASV cultures. In contrast, we only detected TNFα, TNFβ, and MIP-1β production by R848-treated pDCs. More surprisingly, IL-6, IL-15, and MCP-1 were produced in higher amounts by LASV-infected than MOPV-infected pDCs.

We next conducted an unbiased evaluation of pDC activation by transcriptomic analysis (Appendix A). Genes regulated during MOPV or LASV infection are presented in Figure 3b,c. Globally, genes up (or down) regulated during MOPV infection were also up (or down) regulated during LASV infection. However, the fold-changes in gene expression were higher for MOPV-infected pDCs than LASV-infected pDCs. This result underlines the higher immunogenicity of MOPV than LASV.

Many genes identified using this technique are linked to the immune response. The most highly represented families were the JAK/STAT pathway (SOCS1, SOCS3, ASB2, CISH1, JAK3, SPSB1, and BCL3), IL1β signalling (ILB, IL1R1, IL1R, MEFV) and CCL3 (CCL3, CCL3L1, and CCL3L3). Many identified genes in the JAK/STAT pathway are ISGs involved in the negative feedback loop of the pathway [14]. These genes were more highly expressed in MOPV-infected than LASV-infected pDCs and could be the consequence of the prior activation of the JAK/STAT pathway. Genes from the IL1β and CCL3 pathways were overexpressed in MOPV-infected pDCs, but not in LASV-infected pDCs, suggesting that MOPV-induced activation of pDC was not limited to IFN-I production. Overall, these results highlight the better response of pDCs to MOPV than LASV.

## 4. Discussion

We first showed that neither MOPV nor LASV infection of pDCs was productive. However, dendritic cells (DCs) express large amounts of α-dystroglycan, which is the MOPV and LASV receptor. DCs also express the alternative LASV receptors ALIX, TIM-1 and DC-SIGN. Finally, LCMV pseudo-particles with LASV glycoprotein have shown a great affinity for pDCs [9]. Consequently, MOPV and LASV viral particles should be able to enter pDCs, although we do not know which receptors are expressed by human pDC. The viral cycle may have been initiated and aborted. We did not detect the production of MOPV and LASV RNAs in pDCs either. Therefore, if the viral cycle is initiated, it is aborted before the viral replication and transcription steps. As pDCs are highly efficient in pathogen detection, it is also possible that viral components were produced but quickly recycled, preventing their detection with our techniques. Studies on the infection of pDC-like cell lines by LCMV have shown no permissivity to viral particles. However, LCMV infection was possible through cellular contact with infected cells [15]. With our model, we were able to detect MOPV and LASV Z proteins inside pDCs cultured with infected VeroE6 cells. These Z proteins could have three origins: neosynthesis, internalized virions or other type of viral components released by infected VeroE6 such as exosomes. Thus, coculture with infected cells either increased pDC permissivity to MOPV and LASV infection or increased the internalisation of viral component by pDCs. Further investigations using transwell experiments could allow to determine whether direct cell-cell contacts are needed for viral transmission to pDC. Depending on its origin, the Z protein could induce or elude different kinds of recognition pathways, which could affect pDC activation. We also showed that coculture with pDCs lowered the MOPV and LASV infection of VeroE6 cells. Production of IFN-I by pDCs could explain the control of VeroE6 infection, suggesting that pDCs respond to both MOPV- and LASV-infected VeroE6.

As pDCs are specialized in IFN-I production, we quantified their IFN-I response to MOPV and LASV. pDCs produced IFN-I in response to MOPV and LASV virions. However, pDC response to MOPV seemed to be more stable over time. This difference may rely on stronger activation by MOPV, allowing a longer response. It could also be explained by the expression of LASV proteins, which have immunosuppressive properties and may shut down the pDC response [12]. Specifically, LASV NP has been shown to have major immunosuppressive properties in myeloid dendritic cells [11]. However, we did not have the opportunity to evaluate the expression of NP in VeroE6-pDC cocultures. pDCs express a wide range of cellular sensors that can detect viral components. It may be informative to identify which pathways are responsible for MOPV and LASV sensing. Some LCMV strains, which are not detected by TLR7, do not induce IFN-I production by pDCs, suggesting that the pDC response to LCMV is TLR7 dependent [7]. However, in vitro, RIG-I is also able to detect LASV and induces IFN-I production [16]. MOPV and LASV may be detected by different sensors, influencing the pDC response.

To improve our understanding of pDC activation during MOPV and LASV infection, we quantified the cytokines produced by pDCs. IFN-I production by MOPV- and LASV-infected pDCs was confirmed. However, the absence of TNFα, TNFβ, and MIP-1β production suggests that MOPV- and LASV-infection did not lead to the complete activation of pDCs, in contrast to that induced by the TLR7 and TLR8 ligand R848. Unexpectedly, we showed that IL-6, IL-15, and MCP-1 were produced in higher amounts by LASV-infected than MOPV-infected pDCs. These three cytokines were previously identified in humans or NHPs with VHF. IL-6 was associated with severe cases of LASV and Ebola virus infection [6,17]. IL-15 and MCP-1 were also associated with a bad prognosis during the Crimean–Congo haemorrhagic fever [18]. In pDCs, pro-inflammatory cytokines appear to be induced through IRF5 and NFκB activation [19]. pDCs could be involved in the initiation of the pro-inflammatory context associated with severe LF, as the pDC response occurs very early. Transcriptomic analysis of MOPV- and LASV-infected pDCs suggested a comparable pattern of activation of pDCs between both viruses. However, MOPV infection had a greater impact on the pDC transcriptomic state compared to LASV infection. Therefore, MOPV seems to be a stronger stimulus than LASV for pDC activation.

Our results show that primary human pDCs are not productively infected by either MOPV or LASV. However, in the presence of infected cells, viral proteins could be detected in pDCs. MOPV-infected pDCs were globally activated, and rapidly and durably produced IFN-I. In contrast, LASV-infected pDCs were less activated, showed shorter IFN-I production, and produced pro-inflammatory cytokines. Further investigation will be necessary to identify the cell sensors involved in the response to LASV and MOPV and determine whether viral proteins are responsible for these different responses. The role of pDC in vivo could be critical, as IFN-I has been correlated with survival to LF. IFN-I produced by pDCs would have direct antiviral effects as well as induce better activation of other immune cells. Therefore, the strength of pDC activation during the very first steps of infection could modulate the global immune response to LASV.

## Figures and Tables

**Figure 1 viruses-11-00287-f001:**
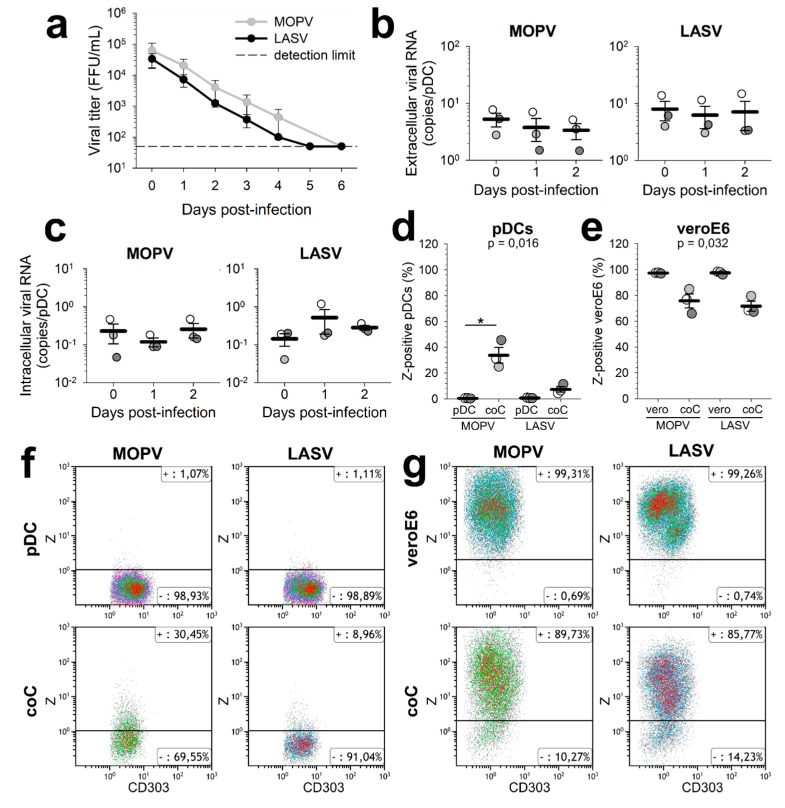
pDC infection by MOPV and LASV. (**a**) pDCs were infected with MOPV or LASV (MOI = 0.1) and infectious particles in the culture medium quantified. (**b**,**c**) pDCs were infected with MOPV or LASV (MOI = 0.1) for 1 h (day 0), 1 day, or 2 days. Viral RNA in the culture medium (**b**) or inside the cells (**c**) was quantified by RT-qPCR. (**d**–**g**) VeroE6 cells were infected with MOPV-Zflag or LASV-Zflag (MOI = 0.3). After 24 h, pDCs were added to the cells, or infected with MOPV-Zflag or LASV-Zflag (MOI = 0.1). 24 h later, cells were stained for phenotypic markers and the Z protein. Conditions were: infected VeroE6 ("veroE6"), VeroE6 cultured with pDCs ("coC"), and infected pDCs (“pDC”). Z-positive pDCs (**d**–**f**) and VeroE6 cells (**e**–**g**) were quantified by flow cytometry. All data are presented as the mean and standard error of mean (SEM) of three independent experiments. ANOVA on Ranks followed by pairwise comparisons (Tukey test) were performed. Differences are significant for *p* < 0.05. When significant, *P* values of the ANOVA are indicated on the graph. Significant pairwise comparisons are indicated by a star (*).

**Figure 2 viruses-11-00287-f002:**
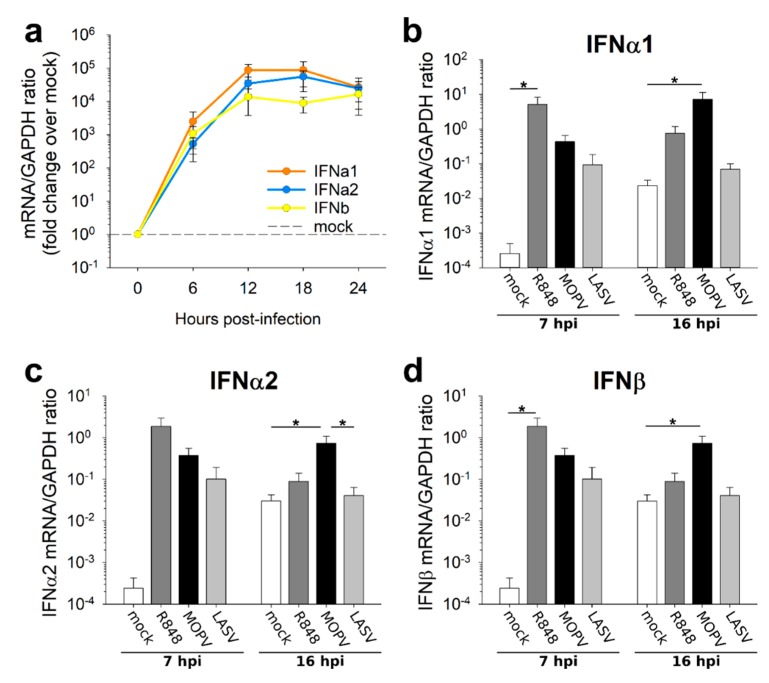
IFN-I production in LASV-infected pDCs is less long-lasting than that of MOPV-infected pDCs. (**a**) pDCs were infected with MOPV (MOI = 2). Every 6 h, from 0 to 24 hpi, IFN-I mRNA was quantified by RT-qPCR. Data are presented as the fold change in the mRNA/GAPDH ratio in MOPV-infected pDCs relative to uninfected pDCs. (**b**–**d**) pDCs were cultured for 7 h or 16 h in culture medium (mock), R848 (1 µg/mL), MOPV, or LASV (MOI = 2). IFNα1 (**b**), IFNα2 (**c**) and IFNβ (**d**) mRNAs were quantified by RT-qPCR. Data shown are the means and SEM of three (**a**), four (**b**–**d** – 7 hpi), or seven (**b**–**d** – 16 hpi) independent experiments. ANOVA on Ranks followed by pairwise comparisons (Tukey test) were performed. Differences are significant for *p* < 0.05. Significant pairwise comparisons are indicated by a star (*).

**Figure 3 viruses-11-00287-f003:**
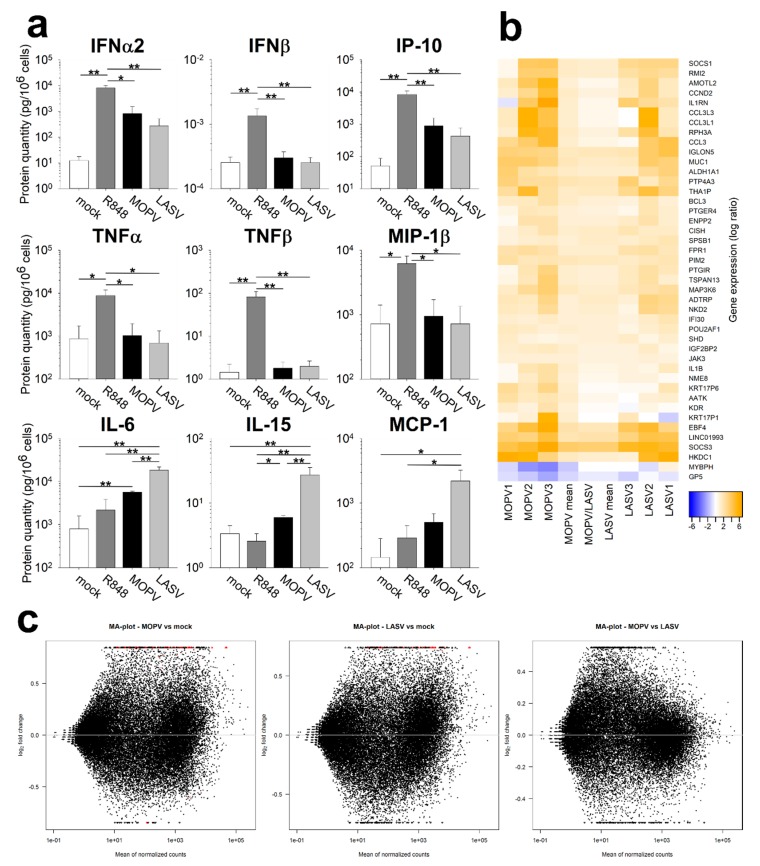
MOPV- and LASV-infected pDCs show different patterns of activation. (**a**) pDCs were cultured for 16 h with culture medium (mock), R848 (1 µg/mL), MOPV, or LASV (MOI = 2). Protein levels were quantified using the Luminex assay. Data are presented as the means and SEM of five independent experiments. Wilcoxon tests were performed, and differences are significant for *p* < 0.05 (*) or *p* < 0.01 (**). (**b**,**c**) pDCs were cultured for 12 h in culture medium (mock), MOPV, or LASV (MOI = 1). Cellular mRNA from three independent experiments was quantified by poly-A amplification and next-generation sequencing. (**b**) Data show the differential expression of genes in MOPV relative to LASV (MOPV/LASV) infected cells or in MOPV or LASV infected cells relative to mock (1, 2, 3, and mean). Genes shown in this figure displayed significant differences of expression (adjusted *p* < 0.05). (**c**) MA plots for all pairwise comparison of data sets (MOPV/mock, LASV/mock and MOPV/LASV). Red dots indicate significantly different genes between the two conditions. Triangles correspond to features having a too low/high fold change to be displayed on the plot.

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
