# Peer review of "Non-Pathogenic Mopeia Virus Induces More Robust Activation of Plasmacytoid Dendritic Cells than Lassa Virus"

_viruses, 2019, doi:10.3390/v11030287_

Round 1
Reviewer 1 Report
In the present interesting study, Schaeffer and colleagues investigate the innate response in plasmocytoid dendritic cells (pDC) against the highly pathogenic Lassa virus (LASV) and the closely related non-pathogenic Mopeia virus (MPOV). The Old World arenavirus LASV is an emerging RNA virus associated with severe human disease and merits significant attention as important public health problem. The lack of a licensed vaccine and the limited therapeutic options make the development of novel treatment strategies against LASV an urgent need. A hallmark of fatal LASV infection in humans is marked immunosuppression, resulting in unchecked viral multiplication, shock, and death. Seminal studies by Dr. Baizes laboratory in non-human primates demonstrated that survival of LASV infection correlated with an early peak of type-1 interferon (IFN-I) production. In experimental infection of the prototypic Old World arenavirus lymphocytic choriomeningitis virus (LCMV) in the mouse, pDCs were found to be important sources of IFN-I early in viral infection. However, the role of pDC in early innate detection of LASV remained unclear.
Using a suitable in vitro model, the authors show that neither free LASV nor free MOPV can productively infect primary human pDC. This is in contrast to the murine LCMV model, where productive infection of pDC with free virus is observed in vitro and emphasizes the importance of using the relevant virus in human cells. When co-cultured with infected cells, LASV and MOPV were able to spread into pDC, which is a likely scenario for the in vivo infection. When exposed to LASV and MPOV, pDC up-regulated IFN-I expression without undergoing productive infection. This is an important observation, in line with findings in the LCMV model, where infected and uninfected pDC produce IFN-I. Using transcriptome profiling and Luminex assay, the authors show that LASV and MOPV differentially induce expression of IFN-I, cytokines, and chemokines in pDCs, which correlated with their different disease potential. The study addresses a timely and interesting issue in LASV pathogenesis. The experimentation is technically sound and the data are presented in a straightforward manner. The conclusions are largely supported by the data. The study provides evidence for a novel and interesting mechanism by which LASV and MOPV differentially activate pDCs. This likely accounts for some of the crucial differences in early innate immunity to LASV and MOPV that account for the strikingly distinct disease outcome and will be of interest to the field. There are no major weaknesses and I have only some small comments that the authors may consider.
Specific comments:
1. Line 35: LCMV is not a highly pathogenic arenavirus. In immunocompetent humans, LCMV is a mild pathogen that can cause aseptic meningitis/encephalitis, but has only low mortality. The virus is only dangerous for immunocompromised patients, during pregnancy, and in newborns.
2. Line 41: A hallmark of fatal LASV infection in humans is marked immunosuppression, resulting in unchecked viral multiplication, shock, and death. Survivors develop a timely adaptive immune response that controls and finally eliminates the virus. Please rephrase.
3. The authors convincingly show that there is negligible infection of free LASV and MPOV in human pDC, whereas in co-cultures, the virus can spread to pDC as assessed by detection of the Z protein. Do the authors think that the Z signal detected in pDC in FACS in Fig. 1F stems from transmitted virus or is the consequence of active viral replication? Please discuss briefly. In a follow-up, transwell experiments could be performed to see of direct cell-cell contacts are needed for viral transmission.
4. The reduction of viral load in co-cultered VeroE6 cells in Fig. 1E is interesting. VeroE6 cells cannot produce IFN-I, but are capable of responding to exogenous IFN-I. Is the idea here that IFN-I produced by the co-cultured pDC exerts some control over the virus in VeroE6 cells? Please discuss.
5. Line 201. In the study by Macal et al., 2012 (reference 9), murine pDC cultures ex vivo were found to be susceptible to infection with LCMV. In virus overlay assays performed in that study on extracts of mouse pDC, a virus-binding protein was detected, but its identity remained elusive and I am not aware that it has ever been verified to be dystroglycan. Regarding the present study, is it known if human pDC express functional DG or any of the other candidate receptors like Axl, TIM-1, or DC-SIGN? Please discuss briefly.
6. Line 217, the authors state: “It could also be explained by the expression of LASV proteins, which have immunosuppressive properties and may shut down the pDC response”. In a recent elegant study, the authors clearly demonstrated that the NP of LASV is the major factor responsible for suppression of the host cell’s IFN-I response, whereas Z makes only a minor contribution (reference 11). Have the authors looked at the NP expression levels in LASV and MOPV infected pDC obtained in co-culture with VeroE6?
Minor points:
1. Line 31: West with capital W
2. Line 34: Old World without hyphen.
3. Line 125: VeroE6 with capital V.
4. Line 146 and 149: transcription of the indicated genes was induced or up-regulated. The term “overexpressed” seems in my opinion not appropriate for ISGs and their regulation.
5. Line 200: neither MOPV nor LASV.
6. Line 234: This sentence is difficult to understand, please re-phrase.
7. In the figures, sub-figures are labeled with capital letters, whereas lower case is used in the legend. Please adjust.
Author Response
Manuscript viruses-445090: Response to reviewer 1
General Comments
The manuscript was modified to fit the reviewer’s remarks. In Figure 1, P values of significant ANOVA were added on the graphs (Figure 1d-e). Figure 2 was reorganized as requested by reviewer 2, so that IFN-I qPCR from 7hpi and 16hpi are on the same graph. Figure 3 was completed with MA-plot for the three pairwise comparison of the transcriptomic analysis. Figure S2 was added to respond to reviewer 2 comment on the Luminex analysis. Figure S3 was added to bring additional data on the transcriptomic analysis, as requested by reviewer 2. Introduction was corrected as requested by reviewer 1. Material and Methods section for the transcriptomic analysis was completed according to review 2 demands. Results and discussion were completed to answer reviewer 1 remarks.
Reviewer 1
Specific comments:
1. Line 35: LCMV is not a highly pathogenic arenavirus. In immunocompetent humans, LCMV is a mild pathogen that can cause aseptic meningitis/encephalitis, but has only low mortality. The virus is only dangerous for immunocompromised patients, during pregnancy, and in newborns.
We have removed LCMV from the sentence:
L34-35: This phylogenetic group includes highly pathogenic viruses, such as LASV or Lujo virus and non-pathogenic viruses, such as Mopeia virus (MOPV).
2. Line 41: A hallmark of fatal LASV infection in humans is marked immunosuppression, resulting in unchecked viral multiplication, shock, and death. Survivors develop a timely adaptive immune response that controls and finally eliminates the virus. Please rephrase.
L40-41: Fatal LASV infection in human is associated with immunosuppression, leading to uncontrolled viral replication and severe symptoms.
3. The authors convincingly show that there is negligible infection of free LASV and MPOV in human pDC, whereas in co-cultures, the virus can spread to pDC as assessed by detection of the Z protein. Do the authors think that the Z signal detected in pDC in FACS in Fig. 1F stems from transmitted virus or is the consequence of active viral replication? Please discuss briefly. In a follow-up, transwell experiments could be performed to see of direct cell-cell contacts are needed for viral transmission.
L137-139: Z proteins detected in pDCs could either come from neosynthesis or internalization of existing viral components.
L230-233: These Z proteins could have three origins: neosynthesis, internalized virions or other type of viral components released by infected VeroE6 such as exosomes. Thus, coculture with infected cells either increased pDC permissivity to MOPV and LASV infection or increased the internalisation of viral component by pDCs.
L233-236: Further investigations using transwell experiments could allow to determine whether direct cell-cell contacts are needed for viral transmission to pDC. Depending on its origin, the Z protein could induce or elude different kinds of recognition pathways, which could affect pDC activation.
4. The reduction of viral load in co-cultered VeroE6 cells in Fig. 1E is interesting. VeroE6 cells cannot produce IFN-I, but are capable of responding to exogenous IFN-I. Is the idea here that IFN-I produced by the co-cultured pDC exerts some control over the virus in VeroE6 cells? Please discuss.
L141-143: VeroE6 cells are unable to produce IFN-I, but are susceptible to exogenous IFN-I, while pDCs are specialized in the production of IFN-I in response to pathogens. Thus, the control of VeroE6 infection observed in VeroE6-pDC cocultures could result from the production of IFN-I by pDCs.
L236-238: We also showed that coculture with pDCs lowered MOPV and LASV infection of VeroE6 cells. Production of IFN-I by pDCs could explain the control of VeroE6 infection, suggesting that pDCs respond to both MOPV- and LASV-infected VeroE6.
5. Line 201. In the study by Macal et al., 2012 (reference 9), murine pDC cultures ex vivo were found to be susceptible to infection with LCMV. In virus overlay assays performed in that study on extracts of mouse pDC, a virus-binding protein was detected, but its identity remained elusive and I am not aware that it has ever been verified to be dystroglycan. Regarding the present study, is it known if human pDC express functional DG or any of the other candidate receptors like Axl, TIM-1, or DC-SIGN? Please discuss briefly.
We have changed pDC for DC, as the expression of α-dystroglycan is well known in these cells and have added a sentence:
L218-223: However, DCs express large amounts of α-dystroglycan, which is the MOPV and LASV receptor. DC also express alternative LASV receptors ALIX, TIM-1 and DC-SIGN. Finally, LCMV pseudo-particles with LASV glycoprotein have shown a great affinity for pDCs [9]. Consequently, MOPV and LASV viral particles should be able to enter pDCs, although we do not know which receptors are expressed by human pDC.
6. Line 217, the authors state: “It could also be explained by the expression of LASV proteins, which have immunosuppressive properties and may shut down the pDC response”. In a recent elegant study, the authors clearly demonstrated that the NP of LASV is the major factor responsible for suppression of the host cell’s IFN-I response, whereas Z makes only a minor contribution (reference 11). Have the authors looked at the NP expression levels in LASV and MOPV infected pDC obtained in co-culture with VeroE6?
We have added a sentence:
L243-246: Specifically, LASV NP has been shown to have major immunosuppressive properties in myeloid dendritic cells [11]. However, we did not have the opportunity to evaluate the expression of NP in VeroE6-pDCs cocultures.
Minor points: all minor points were corrected as requested by the reviewer
1. Line 31: West with capital W
2. Line 34: Old World without hyphen.
3. VeroE6 with capital V.
4. Line 162 and 164: transcription of the indicated genes was induced or up-regulated. The term “overexpressed” seems in my opinion not appropriate for ISGs and their regulation.
5. Line 218: neither MOPV nor LASV.
6. Line 234: This sentence is difficult to understand, please re-phrase.
L264-265: However, MOPV infection had a greater impact on pDC transcriptomic state compared to LASV infection.
7. In the figures, sub-figures are labelled with capital letters, whereas lower case is used in the legend. Please adjust.
Sub-figures and legends are now labelled with lower cases.

Reviewer 2 Report
In this manuscript, Schaeffer and colleagues compare the pDC response following infection with the highly pathogenic Lassa virus (LASV) or the closely related non-pathogenic Mopeia virus (MOPV). Their studies demonstrate that while both viruses do not replicate in pDCs, they induce a differential response. Specifically, infection of pDC with MOPV results in the induction of type 1 IFN response that is sustained 16 hrs post infection. In contrast, LASV induces a short-lived response that wanes by 16 hours. Additional analysis of secreted immune factors confirms the reduced production of type I IFN and additional differences in secretion of a few cytokines/chemokines. Finally, the authors carry out gene expression analyses to gain a more global insight into the differential response of pDC to these two viruses.
Overall, these studies provide novel insight into the basis of differential pathogenicity of thestse two viruses. The manuscript could however benefit from additional clarifications, better data visualization, and greater inclusion of the gene expression data generated. Specific concerns and recommendations are outlined below:
1. Figure 1: the authors state that the levels of Z protein expression in lower in Vero cells co-cultured with pDC compared to those that are not. This observation makes sense given that immune mediators, notably tye 1 IFN, produced by the pDCs could inhibit viral replication within Vero cells. Since the authors have clearly carried out the experiment in multiple replicates, they should show the results of the statistical analysis in panel E.
2. Figure 2: rather than separate bar graphs for 7 and 16 hours, it would be far more informative to plot the expression fof each of the type I IFN genes at each of the time points side by side. This will help the reader appreciate the enhances response with MOPV and the flat response with LASV. As presented, it is not clear if the response at 7 hour is statistically greater than that of the Mock. Moreover, the authors should use different statistical test, e.g. a way way ANOVA with multiple comparison correction.
3. Figure 3A: only 9 of 50 factors are shown in panel A. Were the levels of the other 41 factors not changed with infection? A HM of all the values could help provide a reader with a global understanding of the response of pDC to MOPV and LASV
4. Figure 3B: This heatmap is rather obtuse and difficult to interpret. Does it show all the differentially expressed genes (DEG)? Or a subset? If the latter, how was this subset chosen? Similarly the enrichment data in supplemental figure 2 is meaningless without context: how many DEG were used? How many DEGs mapped to a give GO term? Are the enrichment p values corrected for false discover?
To mitigate these concerns, I strongly urge the authors to show more of the gene expression data in a separate figure. Specifically, the authors should provide the total number of differentially expressed genes (DEG) detected after LASV and MOPV Infection respective. It would be extremely helpful if they provided a Venn diagram so that the reader can see differences and similarities in the two transcriptional responses. The authors make a general statement that DEG detected after MOPV infection show a greater fold change than following LASV infection. This cannot be gleaned from the heatmap provided. Instead the authors should take the common DEG and show their expression in a violin plot or a scatter plot to clearly outline differences in fold induction/normalized transcript counts.
Instead of showing a large number of GO terms, the authors should focus on those terms with highly significant FDR corrected p value and to which a large number of genes enriched.
The methods section on the transcriptional analysis requires a lot more details: how were the libraries generated (which kit)? What was the average number of reads/sample? How was the quality of the reads checked? Were reads trimmed? How? What genome was the alignments against? what was the number of mismatches allowed? How were the reads normalized? What was the definition of a differentially expressed genes? The authors should also provide a PCA or a hierarchial clustering to show the intra-group variability.
Author Response
Manuscript viruses-445090: Response to reviewer 2
General Comments
The manuscript was modified to fit the reviewer’s remarks. In Figure 1, P values of significant ANOVA were added on the graphs (Figure 1d-e). Figure 2 was reorganized as requested by reviewer 2, so that IFN-I qPCR from 7hpi and 16hpi are on the same graph. Figure 3 was completed with MA-plot for the three pairwise comparison of the transcriptomic analysis. Figure S2 was added to respond to reviewer 2 comment on the Luminex analysis. Figure S3 was added to bring additional data on the transcriptomic analysis, as requested by reviewer 2. Introduction was corrected as requested by reviewer 1. Material and Methods section for the transcriptomic analysis was completed according to review 2 demands. Results and discussion were completed to answer reviewer 1 remarks.
Reviewer 2
1. Figure 1: the authors state that the levels of Z protein expression in lower in Vero cells co-cultured with pDC compared to those that are not. This observation makes sense given that immune mediators, notably type 1 IFN, produced by the pDCs could inhibit viral replication within Vero cells. Since the authors have clearly carried out the experiment in multiple replicates, they should show the results of the statistical analysis in panel E.
P values of the ANOVA tests were added on the graphs (when significant). Stars on the graphs indicate significant pairwise comparison.
2. Figure 2: rather than separate bar graphs for 7 and 16 hours, it would be far more informative to plot the expression for each of the type I IFN genes at each of the time points side by side. This will help the reader appreciate the enhances response with MOPV and the flat response with LASV. As presented, it is not clear if the response at 7 hours is statistically greater than that of the Mock. Moreover, the authors should use different statistical test, e.g. a way way ANOVA with multiple comparison correction.
Figure 2 was modified to show 7hpi and 16hpi on the same graph. ANOVA on rank followed by pairwise comparisons (Tukey test) were performed. Significant pairwise comparison are now indicated by (*).
3. Figure 3A: only 9 of 50 factors are shown in panel A. Were the levels of the other 41 factors not changed with infection? A HM of all the values could help provide a reader with a global understanding of the response of pDC to MOPV and LASV
For the Luminex analysis, only factors showing significant differences between at least two of the conditions are showed in Figure 3a. A table was added in supplementary data (Figure S2) showing all the factors tested, those for which the quantities were too low to be detected, those without differences between the conditions and those with significant differences.
4. Figure 3B: This heatmap is rather obtuse and difficult to interpret. Does it show all the differentially expressed genes (DEG)? Or a subset? If the latter, how was this subset chosen? Similarly the enrichment data in supplemental figure 2 is meaningless without context: how many DEG were used? How many DEGs mapped to a give GO term? Are the enrichment p values corrected for false discover?
The heatmap on Figure 3b shows all genes which were differentially regulated in on of the pairwise comparison. Genes are differentially selected for p<0.05. Globally, the number of DEG was quite low.
To mitigate these concerns, I strongly urge the authors to show more of the gene expression data in a separate figure. Specifically, the authors should provide the total number of differentially expressed genes (DEG) detected after LASV and MOPV Infection respective. It would be extremely helpful if they provided a Venn diagram so that the reader can see differences and similarities in the two transcriptional responses. The authors make a general statement that DEG detected after MOPV infection show a greater fold change than following LASV infection. This cannot be gleaned from the heatmap provided. Instead the authors should take the common DEG and show their expression in a violin plot or a scatter plot to clearly outline differences in fold induction/normalized transcript counts.
We added Figure S3 to provide more information on our transcriptomic approach. This figure contains a density plot for all 9 samples, two graphs of the Principal Component Analysis, count distribution before and after normalisation, and the number of DEG for all pairwise comparison. In Fig3c, we added MA-plots for the 3 pairwise comparison (with DEG in red).
Instead of showing a large number of GO terms, the authors should focus on those terms with highly significant FDR corrected p value and to which a large number of genes enriched.
The methods section on the transcriptional analysis requires a lot more details: how were the libraries generated (which kit)? What was the average number of reads/sample? How was the quality of the reads checked? Were reads trimmed? How? What genome was the alignments against? what was the number of mismatches allowed? How were the reads normalized? What was the definition of a differentially expressed genes? The authors should also provide a PCA or a hierarchial clustering to show the intra-group variability.
M&M section was completed as requested.
L97-111: pDCs were infected for 12 h at a MOI = 1 with LASV, MOPV, or remained uninfected. Cellular RNA was purified using the RNeasy kit (Qiagen), followed by DNAse I (Qiagen) and DNAse (Ambion) digestion. Sequencing were performed by ViroScan3D (Lyon, France). RNA quality was checked using QuantiFluor RNA System (Promega) and RNA 6000 Pico Kit (Agilent). cDNA were synthesized using random priming of poly-A RNAs (NEXTFLEX Rapid Directional RNA-Seq Library Prep Kit, PerkinElmer). Single-end, 75-bp ead-length NextSeq 500 High output sequencing of the cDNA library was performed. After demultiplexing and trimming of the adaptors (with Bcl2fastq), 30 million reads per sample were obtained. Sequencing quality was assessed for each sample (before and after mapping) using FastQC. Reads were aligned on human genome (Human GRCh38.p7, from ENSEMBL) using STAR and a maximum mismatch rate of 5%. Reads aligned on each gene were counted using the module Feature count. Statistical analysis of the read counts (quality checks, normalisation using scaling factors, Fold change and P values calculation) were performed using the R package SARTools (DESeq2) [13]. Genes were differentially expressed for p<0.05. Heatmap were made with R (package heatmap2), using genes with differential expression for at least one pairwise comparison.
